



# Linking marine phytoplankton emissions, meteorological processes and downwind particle properties with FLEXPART

Kevin J. Sanchez[1,2], Bo Zhang[3], Hongyu Liu[3], Georges Saliba[4], Chia-Li Chen[4], Savannah L. Lewis[4], Lynn M. Russell[4], Michael A. Shook[2], Ewan C. Crosbie[2,5], Luke D. Ziemba[2], Matthew D. Brown[2,5], Taylor J. Shingler[2], Claire E. Robinson[2,5], Elizabeth B. Wiggins[1,2], Kenneth L. Thornhill[2,5], Edward L. Winstead[2,5], Carolyn Jordan[2,3], Patricia K. Quinn[6], Timothy S. Bates[6,7], Jack Porter[9], Thomas G. Bell[8,9], Eric S. Saltzman[9], Michael J. Behrenfeld[10], and Richard H. Moore[2]

[1]NASA Postdoctoral Program, Universities Space Research Association, Columbia, MD
[2]NASA Langley Research Center, Hampton, VA
[3]National Institute of Aerospace, Hampton, VA
[4]Scripps Institution of Oceanography, University of California San Diego, La Jolla, CA
[5]Science Systems and Applications, Inc., Hampton, VA
[6]Pacific Marine Environmental Laboratory, NOAA, Seattle, WA, USA
[7]Joint Institute for the Study of the Atmosphere and Ocean (JISAO), University of Washington, Seattle, WA, USA
[8]Plymouth Marine Laboratory, Prospect Place, Plymouth, United Kingdom
[9]The Department of Earth System Science, University of California, Irvine, CA, USA
[10]Oregon State University, Corvallis, OR

*Correspondence to*: Kevin J. Sanchez (kevin.j.sanchez@nasa.gov) and Richard H. Moore (richard.h.moore@nasa.gov)

**Abstract.** Marine biogenic particle contributions to atmospheric aerosol concentrations are not well understood though they are important for determining cloud optical and cloud nucleating properties. Here we examine the relationship between marine aerosol measurements with satellite and model fields of ocean biology and meteorological variables during the North Atlantic Aerosols and Marine Ecosystems Study (NAAMES). NAAMES consisted of four field campaigns between November 2015 and April 2018 that aligned with the four major phases of the annual phytoplankton bloom cycle. The FLEXPART Lagrangian particle dispersion model is used to connect these variables spatiotemporally to ship-based aerosol and dimethyl sulphide (DMS) observations. We find that correlations between some aerosol measurements with satellite measured and modelled variables increase with increasing trajectory length, indicating biological and meteorological processes over the air mass history are influential to measured particle properties and that using only spatially coincident data would miss correlative connections that are lagged in time. In particular, the marine non-refractory organic aerosol mass correlates with modelled marine net primary production when weighted by 5-day air mass trajectory residence time (r = 0.62). This result indicates non-refractory organic aerosol mass is influenced by biogenic volatile organic compound (VOC) emissions from photosynthesis by phytoplankton stocks during advection into the region. This is further supported by the correlation of non-refractory organic mass with 2-day residence-time-weighted chlorophyll-a (r = 0.39), a proxy for phytoplankton abundance, and 5-day residence-time-weighted downward shortwave forcing (r = 0.58), a requirement for photosynthesis. In contrast, DMS (formed through



biological processes in the seawater) and primary marine aerosol (PMA) concentrations showed better correlations to explanatory biological and meteorological variables weighted with shorter air mass residence times, which reflects their localized origin as primary emissions. Aerosol submicron number and mass negatively correlate with sea surface wind speed. The negative correlation is attributed to enhanced PMA concentrations under higher wind speed conditions. We hypothesized

that the elevated total particle surface area associated with high PMA concentrations leads to enhanced rates of VOC condensation onto PMA. Given the high deposition velocity of PMA, relative to submicron aerosol, PMA can limit the accumulation of secondary aerosol mass. This study provides observational evidence for connections between marine aerosols and underlying ocean biology through complex secondary formation processes, emphasizing the need to consider airmass history in future analyses.

**1 Introduction**

Marine environments are sensitive to aerosol particle loading because particles can act as cloud condensation nuclei (CCN) on which cloud droplets form. The number concentration of cloud droplets can influence cloud optical properties and therefore affect the impact of clouds on climate. Measurements over the ocean are scarce and have historically been concentrated over relatively few areas and for short time periods aligning with episodic intensive ship or aircraft campaigns. Satellite

measurements are crucial to filling this void in marine boundary layer (MBL) measurements, but satellite optical measurements of particles are disproportionally weighted by the largest optically-active (>300 nm diameter) particles (Hasekamp et al., 2019). An unresolved challenge is to relate these remote sensing measurements to the particle number, size distribution, and chemical composition that are known to drive variability in cloud properties. Since ocean-emitted volatile compounds and particles can control the number, size and composition of marine aerosols (Brooks and Thornton, 2018), satellite measurements of ocean

biomass are investigated as a possible proxy for marine particle properties.

In the absence of advected continental or anthropogenic pollution, the main source of MBL particles is sea salt from primary marine aerosol (PMA) and marine biogenic volatile organic compounds (VOCs) emissions that can oxidize and condense to form secondary aerosol mass (Bates et al., 1998a; Covert et al., 1992; Frossard et al., 2014b; Murphy et al., 1998; Quinn et al., 2000, 2014; Rinaldi et al., 2010; Sievering et al., 1992b, 1999; Warren and Seinfeld, 1985). Clear seasonal trends have been

identified between ocean chlorophyll-a, a frequently used proxy for marine biomass, and marine biogenic particle production (Odowd et al., 1997; Ovadnevaite et al., 2014; Van Pinxteren et al., 2017; Saliba et al. submitted; Shaw, 1983). This observed seasonal trend results from the marine phytoplankton cycle, which is driven by ocean mixed layer deepening in the winter months, increasing nutrients at the surface and decoupling phytoplankton from predators, followed by increased sunlight in the spring, enhancing photosynthetic primary productivity; they together produce the annually-occurring North Atlantic

phytoplankton bloom (Behrenfeld, 2010; Behrenfeld and Boss, 2018; Boss and Behrenfeld, 2010). The bloom ends when rates of loss processes equal or exceed rates of growth and is driven by shoaling of the mixed layer in the spring, which cuts off the surface layer from deep water nutrients (Behrenfeld and Boss, 2018).



One of the most studied biogenically-produced marine aerosol components is non-sea-salt sulfate (nssSO4) because of its role as a proposed link to a biological thermostat on clouds (Charlson et al., 1987; Shaw, 1983), a notion that has been called into question (Quinn and Bates, 2011). Marine nssSO4 is formed from the oxidation products of dimethyl sulfide (DMS), a VOC derived from marine ecosystem processes (Yoch, 2002). The phytoplankton cycle leads to a seasonal variation in DMS-derived sulfate aerosol mass, with higher sulfate concentrations during bloom periods (Bates et al., 1998a; Bell et al. in prep; Frossard et al., 2014b; Park et al., 2017; Quinn et al., 2019; Sanchez et al., 2018). DMS-derived sufuric acid can either increase the size of pre-existing MBL particles by adding to the aerosol sulfate mass through condensation or can nucleate new particles in regions where existing particle surface area is deficient. While a seasonal relationship between DMS and marine nssSO4 mass exists, a direct temporal link is often not clear due to the range of chemical pathways affecting the yields of secondary aerosol formation and influence of entrainment on the formation of new particles (Andreae and Crutzen, 1997; Ayers et al., 1997; Bates et al., 1998a; Chen and Jang, 2012; Clarke et al., 1999, 2013; O'Connor et al., 2008; Reus et al., 2000; Sanchez et al., 2018; Veres et al., 2020). The formation rate of the DMS oxidation products, methane sulfonic acid, sulfuric acid and hydroperoxymethyl thioformate is a function of temperature and availability of atmospheric oxidants (Seinfeld and Pandis, 2006; Veres et al., 2020). Mixing of DMS across the MBL inversion, typically driven by convective or buoyancy driven transport, can lead to subsequent entrainment of newly formed particles back into the MBL. New particle formation is thought to be favored in the free troposphere due to the colder temperatures and low pre-existing particle surface area (Clarke, 1993; Clarke et al., 2013; Raes et al., 1997; Russell et al., 1998; Sanchez et al., 2018; Seinfeld and Pandis, 2006; Thornton et al., 1997; Yue and Deepak, 1982). Furthermore, sulfate mass can rapidly form onto existing particles through cloud processing, which enhances rates of sulfur dioxide (a DMS oxidation product and precursor to sulfuric acid) condensation onto cloud droplets due to the enhanced surface area provided by the addition of water. Aqueous phase oxidation of the sulfur dioxide results in the formation of low-volatility sulfuric acid, which remains in the particle phase when the cloud dissipates. This shifts some particles from the Aitken- to accumulation-mode and gives rise to the distinctive 'Hoppel minimum' (a minimum in the particle size distribution concentration between the two modes), commonly observed in cloud-processed marine aerosol size distributions (Hoppel et al., 1986).

Secondary organic aerosol mass is formed through the oxidation of biogenic VOCs such as isoprene and monoterpenes (Altieri et al., 2016; Hallquist et al., 2009). At Cape Grim, chlorophyll-a, an indicator for biogenic VOC emissions, is shown to strongly correlate with organic mass (r = 0.85) on a seasonal timescale (Cui et al., 2019). While isoprene and monoterpenes are known precursors for secondary organic particle mass, models indicate previously observed particle yields and estimated air-sea fluxes of isoprene (2%, 13–38 μg m$^{-2}$d$^{-1}$) and monoterpenes (~32%, 0.27–0.78 μg m$^{-2}$d$^{-1}$) (Hu et al., 2013; Lee et al., 2006) are too low to account for the observed MBL organic mass, suggesting that there may be large undiscovered sources (Arnold et al., 2009; Myriokefalitakis et al., 2010). One proposed possible source is secondary organic precursors derived from photosensitized reactions of dissolved organic matter present in the sea surface microlayer (Cui et al., 2019). In addition, organic aerosol mass has been shown to correlate with black carbon (BC) and other continental tracers that indicate a large portion of the mass is not of marine origin (Saliba et al. submitted; Shank et al., 2012). Despite the presence of continental



organic mass from long-range transport and primary OM, marine biogenically emitted VOCs are estimated to account for more than 60% of organic compounds over remote oceans (Brüggemann et al., 2018). Similarly, dual carbon isotope analysis of marine aerosol indicates that 80% of all primary and secondary organic matter is from biogenic origin during non-polluted
conditions (Ceburnis et al., 2011). For the North Atlantic Aerosols and Marine Ecosystems Study (NAAMES) campaigns, Saliba et al. (submitted) noted the measured submicron non-refractory nitrate mass (a product of secondary processes) strongly correlated with the non-refractory organic mass, suggesting much of the non-refractory organic mass is secondary. Even so, there is still a variable portion of marine organic mass that is emitted as PMA. PMA is composed mainly of sea salt and organic aerosol mass, and accounts for a small number of marine particles, but a significant fraction of the total aerosol mass, due to
their large sizes (0.05-10µm) (Grythe et al., 2014). PMA formation is driven by wave breaking and bubble bursting which is primarily controlled by the surface wind speed (de Leeuw et al., 2011; Modini et al., 2015; Thorpe, 1992). While PMA typically accounts for a low fraction of the marine particle CCN concentration (Fossum et al., 2018; Quinn et al., 2017, 2019), a recent modelling study validated with observations suggests that PMA particles regulate secondary particle contributions to CCN over the remote MBL (Fossum et al., 2020). Enhanced condensation of water vapor onto these coarse-mode PMA particles
can reduce in-cloud supersaturations, preventing the activation of smaller Aitken-mode particles. PMA are short lived with the largest PMA having deposition velocities several orders of magnitude greater than accumulation mode particles (Petroff and Zhang, 2010; Pryor and Barthelmie, 2000; Williams et al., 2002), indicating aerosol mass gained through gas-to-particle conversion onto PMA is quickly removed from the atmosphere. During the NAAMES campaigns, PMA contributions to CCN concentrations at 0.1% supersaturation were found to be low for the North Atlantic (averaging 14-31% for all but the late
autumn NAAMES campaign) (Quinn et al., 2019). Similarly, in global models, PMA were shown to account for about 20-40% of CCN in the North Atlantic (Pierce and Adams, 2006; Yu and Luo, 2009). These observations indicate that a substantial fraction is likely from secondary biogenic sources or from continental sources. Meskhidze and Nenes (2006) reported cloud droplet number concentrations were doubled over a phytoplankton bloom, relative to the surrounding area. Several other studies have also shown that ocean productivity and biomass abundance relate to the spatial and temporal variability in MBL
particle and cloud droplet concentrations (McCoy et al., 2015; Meskhidze and Nenes, 2010; Vallina et al., 2006). While sulfate is more hygroscopic than organic aerosol mass, the latter still contributes to the particle size (which lowers the Kelvin effect) and may decrease the surface tension (Frossard et al., 2018; Ovadnevaite et al., 2017). Organics are likely to contribute to CCN concentrations considering organics often account for a significant fraction of marine Aitken- and accumulation-mode particle mass.

Linking aerosol and cloud properties to marine emissions is complicated by the influence of ship emissions and aerosols transported from the continents, including biomass-burning smoke (Coggon et al., 2012; Shank et al., 2012; Yang et al., 2016). Continental pollution and DMS-derived sulfate that is lofted into the free troposphere can lead to long-range transport of particles and subsequent reentrainment of these particles back into the MBL in a different location (Clarke et al., 2013; Dzepina et al., 2015; Korhonen et al., 2008; Quinn et al., 2019; Saliba et al. submitted; Shank et al., 2012). It is also known that DMS
can persist over significant transport distances and contribute to secondary aerosol production in locations that are





spatiotemporally removed from the source region (Mungall et al., 2016). Similarly, Zavarsky et al. (2018) show that measured and calculated isoprene and PMA fluxes positively correlate with up-wind satellite aerosol measurements. Marine particle concentrations have also been shown to negatively correlate with precipitation along the air mass history given the role of precipitation scavenging as a prominent aerosol loss process (Andronache, 2004; Pruppacher and Klett, 1997; Sanchez et al.

in prep, 2020; Vallina et al., 2006). Observations of new particle formation events correspond with precipitation over the air mass history, likely due to a decrease in coarse-mode aerosol concentrations and total particle surface area with precipitation scavenging (Andronache, 2004; Ueda et al., 2016).

Previous literature hints that phytoplankton activity is related to emissions of organic and sulfate aerosol mass precursors (Altieri et al., 2016; Ayers et al., 1997; Bates et al., 1998b; Brüggemann et al., 2018; Ceburnis et al., 2011; Facchini et al.,

2008; Hallquist et al., 2009; Hu et al., 2013; Huang et al., 2018; Mansour et al., 2020; Ovadnevaite et al., 2014; Park et al., 2017; Quinn et al., 2019; Sanchez et al., 2018). These precursors are emitted over a large area at varying rates due to the spatial and temporal variation of marine biological activity (Behrenfeld, 2010; Behrenfeld and Boss, 2018). The MBL residence time and transport need to be accounted for to study the effect of this area source on particle concentrations and composition. The goal of this work is to look for correlative connections between measured aerosol loading and composition with ocean biology,

physical ocean characteristics, and atmospheric boundary layer meteorology to identify the influence of up-wind processes and biology on particulate mass formation. We use the state-of-the-art FLEXPART trajectory model to identify the contribution of continental transport and to evaluate the connection between North Atlantic aerosols and potential explanatory variables weighted by air mass history and boundary layer residence time.

## 2 Methods

Here, we describe the measurements made on-board the *R/V Atlantis* during NAAMES as well as the satellite and model data products that we explore as explanatory variables. Analysis details including filtering criteria are also discussed.

### 2.1 *R/V Atlantis* Measurements during NAAMES

The NAAMES campaigns were conducted on the *R/V Atlantis* over the four major periods of the North Atlantic marine phytoplankton cycle, aligning with the phytoplankton biomass minimum (November 2015, NAAMES1), maximum (May –

June 2016, NAAMES2), and the transitions marked by the decay of biomass (September 2017, NAAMES3) and accumulation of biomass (March 2018, NAAMES4). Grey lines in Figure 1a-d show the ship track for each campaign. A detailed description of each NAAMES campaign can be found in Behrenfeld et al. (2019).

Aerosols were sampled on the forward O2 deck of the *R/V Atlantis* with a temperature-controlled isokinetic inlet approximately 18 m above sea level. Particles were dried in diffusion driers before being measured by instruments. Supermicron particles

were sized with an Aerodynamic Particle Sizer (APS 3321, TSI Inc., St. Paul, MN, size range 0.5–20 µm). Additional aerosol instrumentation was downstream of a 1.0 µm sharp cut cyclone (SCC 2.229, BGI Inc. US) to measure only the submicron aerosol fraction. A Condensation Particle Counter (CPC 3010, TSI Inc., St. Paul, MN) was used to measure particle number



concentrations. Because the SEMS CPC uses liquid to grow the particles large enough to be optically counted, the particle number concentrations measured by the CPC are commonly referred to as condensation nuclei or CN. A Scanning Electrical

Mobility Sizer (SEMS, Model 138, 2002, BMI, Hayward, CA) measured particle size distributions (0.02-0.9 µm diameter) at five-minute intervals, utilizing a CPC (CPC 3025, TSI Inc., St. Paul, MN) to count the number of particles in each size bin of the aerosol size distribution. Particle concentrations for diameters greater than and less than 100 nm ($N_{>100nm}$, $N_{<100nm}$) are derived from merged SEMS and APS particle size distributions. A Single Particle Soot Photometer (SP2, DMT, Boulder, CO) measured refractory black carbon mass concentration. Submicron particles were analyzed with a high-resolution time-of-flight

aerosol mass spectrometer (AMS, Aerodyne Research Inc., Billerica, MA) (DeCarlo et al., 2006) that measures non-refractory inorganic (sulfate, ammonium, nitrate, chloride) and organic components. The AMS does not efficiently measure refractory sea salt particles. The PMA particle number concentration is determined by integrating the fitted coarse-mode of the SEMS and APS merged particle size distribution (Modini et al., 2015; Saliba et al., 2019). Radon was measured with a dual-flow-loop two-filter 103 radon detector (Whittlestone and Zahorowski, 1998). Continuous DMS measurements were made by

atmospheric pressure chemical ionization mass spectrometers (Bell et al., 2013, 2015). One instrument collected air measurements, while the other analyzed gas exiting the seawater equilibrator. Inline chlorophyll-a measurements are calculated from AC-S hyper-spectral spectrophotometer measurements (WET Labs, Inc., Philomath, OR) using the "line height" method which relates the phytoplankton absorption at 676 nm to chlorophyll-a from high pressure liquid chromatography samples collected during NAAMES (Boss et al., 2013). In-line particulate organic carbon is derived from beam attenuation data with

the AC-S spectrometer (Boss et al., 2013).

## 2.2 Satellite Data Products

Several biological parameters are obtained from merged satellite ocean color products derived by the GlobColour project (Maritorena et al., 2010). In this paper, four GlobColour level-3 satellite products, related to phytoplankton biomass, represent biological processes at 1° horizontal resolution. The first is chlorophyll-a, a primary photosynthetic pigment in phytoplankton.

Chlorophyll-a is commonly used as a proxy for the biomass of phytoplankton (Behrenfeld et al., 2016; Lyngsgaard et al., 2017; Meskhidze and Nenes, 2010; Pastor et al., 2013). The GlobColour project has several different chlorophyll-a products that are derived from different methods. In this study we use the chlorophyll-a product derived from the Garver-Siegel-Maritonena ocean color model (Maritorena and Siegel, 2005). This specific chlorophyll-a product is chosen because the daily average product best correlates with in-line chlorophyll-a measurements (r = 0.87, Table S7, Figure S1). Figure 1e-h shows the merged

satellite derived chlorophyll-a concentration over the North Atlantic averaged for each NAAMES campaign. The next biological parameter studied is marine particulate organic carbon, which is an important component in the carbon cycle that forms through photosynthesis and subsequent ecosystem processes. In-line particulate organic carbon also correlates with the satellite product (r = 0.70, Table S7, Figure S1). The absorption coefficient of colored detrital organic materials (CDM) is also used as a parameter. This absorption coefficient is largely associated with the chromophoric dissolved organic matter (CDOM),

which is the fraction of dissolved organic matter that interacts with solar radiation (Nelson and Siegel, 2013). CDOM is also



a photosensitizer in the photolysis of DMS, meaning CDOM generates reactive species upon the absorption of solar radiation that remove sea water DMS through oxidation (Bouillon and Miller, 2004; Brimblecombe and Shooter, 1986; Toole et al., 2003, 2008). The final parameter, the depth of the euphotic zone, represents the depth at which down-welling irradiance is 1% of the value at the surface. This depth roughly characterizes the layer of the ocean that can support net phytoplankton
photosynthesis, but is also a function of chlorophyll-a (Morel et al., 2007).

Even though in-line measurements correlate better with one-day average products (Figure S1), we used eight-day average products for all analyses because of the improved spatial coverage (reduced interference from clouds). Eight-day averages have about ~45 % more spatial coverage than one-day averages and approximately 15 % less coverage than monthly averages (Maritorena et al., 2010). If a grid cell is missing data of satellite derived biological parameters, it is filled by averaging the
surrounding eight grid cells. If all surrounding grid cells have missing data, then the next and previous eight-day averages are averaged together to fill the grid cell. This method sufficiently filled all missing data points.

## 2.3 Modelled Net Primary Production

Net primary production is the formation of organic material through photosynthesis by phytoplankton. This process leads to the emission of biogenic VOCs at the sea surface (Li et al., 2018). In general, net primary production is a function of the
photosynthetically available radiation, the euphotic zone depth, the phytoplankton concentration and the efficiency with which carbon biomass is formed (Silsbe et al., 2016). A number of different models calculate net primary production, but in this study we focus on net primary production derived from the Carbon, Absorption, and Fluorescence Euphotic-resolving (CAFÉ) model, which has been shown to be the most accurate in a recent study (Silsbe et al., 2016). The model products are derived from merged satellite data from the Making Earth Science Data Records for Use in Research Environments (MEaSUREs)
NASA initiative (Vollmer et al., 2011). Figure 1(i-l) shows CAFÉ-modeled net primary production over the North Atlantic averaged for each NAAMES campaign.

## 2.4 GDAS Model Reanalysis Data Products

The Global Data Assimilation System (GDAS, ftp://arlftp.arlhq.noaa.gov/pub/archives/gdas1/) gridded output is used to initialize the Global Forecast System (GFS) model with observations obtained from surface observations, radiosondes, wind
profilers, aircraft, buoys, radar and satellite. Here, we use the 3-hour GDAS 1° resolution horizontal sea surface wind speed, low-level cloud cover, three-hour downward shortwave forcing (DSWF) and six-hour accumulated precipitation over the North Atlantic to identify the state of the MBL up-wind of the *R/V Atlantis*.

## 2.5 FLEXPART Back Trajectories

The FLEXible PARTicle dispersion model (FLEXPART) is a Lagrangian particle dispersion model used to estimate transport
pathways of observed air samples (Owen and Honrath, 2009; Stohl et al., 2005; Zhang et al., 2014). Here, we use the FLEXPART model to connect the *R/V Atlantis* observations (section 2.1) to the explanatory variables discussed in Sections



2.2-2.4 by weighting the explanatory variables by the FLEXPART air mass residence time. For the NAAMES campaigns, 10-day FLEXPART backward simulations are initialized along the path of the *R/V Atlantis* every hour. The GFS and its "Final Analysis" drives all the simulations with 3-hour temporal resolution, 1° horizontal resolution, and 26 vertical levels that cover

depth of the troposphere and extend into the stratosphere. In each simulation, ten thousand passive particle tracers are released at the ship location. The advection and dispersion of the particles are simulated backwards in time. The product is an up-wind spatial distribution of the particle residence times (average time an air parcel stays within a model grid cell). Figure 1(a-d) shows the residence time integrated over all 26 vertical levels for the first 5-days of all the FLEXPART trajectories during clean marine periods (Section 2.6). The residence time in Figure 1(a-d) is normalized by the total residence time of all clean

marine trajectories to show the residence time fraction over each grid cell during each NAAMES campaign. For the remaining analysis in this paper, the vertical structure of the residence time is column integrated over only the vertical levels that are completely or partially within the MBL based on GDAS MBL heights. Remaining vertical levels were excluded from analysis.

## 2.6 Criteria for Clean Marine Conditions

In order to constrain the impact of meteorological and biological parameters on marine particle chemical composition and

concentration, air masses that are influenced by continental and anthropogenic emissions are excluded from the analysis. "Clean Marine" conditions are defined as periods when (1) total particle number concentrations are below $1000\ \mathrm{cm^{-3}}$, (2) black carbon mass is below $50\ \mathrm{ng\ m^{-3}}$ to filter out ship contamination and continental transport (Betha et al., 2017; Saliba et al. submitted), (3) radon, a continental tracer, is below $500\ \mathrm{mBq\ m^{-3}}$, (4) AMS non-refractory organic aerosol mass is less than $0.5\ \mathrm{\mu g\ cm^{-3}}$ as suggested by prior measurements of organic aerosol mass over the marine environment (Russell et al., 2010),

and (5) less than 25% of the five-day FLEXPART back trajectory residence time passes over continents. Figure 2 shows the fraction of the FLEXPART trajectory time over land for 6-hour to 10-day trajectories and indicates the median fraction for the five-day back trajectory is 25%, so half of the 2236 hours of measurements are removed due to this criterion alone. In the end, 557 samples are representative of clean marine conditions. Despite meeting the clean marine criteria, black carbon mass still moderately (r = 0.51) correlates with AMS non-refractory organic and sulfate aerosol mass (Figure S2). The correlation even

holds at significantly lower black carbon mass concentration thresholds, which is similar to previous findings (Huang et al., 2018; Saliba et al. submitted; Shank et al., 2012), signifying long-range transport or ship emissions contribute to organic aerosol mass concentrations even in the cleanest remote marine environments. However, approximately 75% of the variability in the organic and sulfate aerosol mass is still unaccounted for, indicating potential influence from marine biogenic sources.

## 3 Results and Discussion

An example of the MBL column-integrated residence time of individual FLEXPART back trajectories is shown in Figure 3a. This is then normalized and multiplied by the biological (8-day average) or meteorological (3-hour or 6-hour average) explanatory variables (for example, satellite-derived Chlorophyll-a shown in Figure 3b), to obtain a residence-time-weighted value that, when integrated, represents the average value over the back trajectory. An example of the residence-time-weighted


chlorophyll-a is shown in Figure 3c. Higher values in Figure 3c represent regions where chlorophyll-a had the greatest
influence on the airmass intercepted by the ship, due to the corresponding high residence times (Figure 3a) and high
chlorophyll-a concentrations (Figure 3b) in these regions. The equation below describes the explicit calculation of the
residence-time-weighted explanatory variables,

$$\text{Integrated residence-time-weighted explanatory variable} = \frac{\sum_{t=1}^{T}\sum_{lon=1}^{360}\sum_{lat=1}^{180} R_{t,lon,lat} E_{t,lon,lat}}{\sum_{t=1}^{T}\sum_{lon=1}^{360}\sum_{lat=1}^{180} R_{t,lon,lat}}, \qquad (1)$$

where $R_{t,lon,lat}$ and $E_{t,lon,lat}$ are the MBL residence time and explanatory variable values, respectively, at each hour (t), longitude
(lon), and latitude (lat) for a FLEXPART back trajectory with a length of T hours. Weights are applied evenly at all trajectory
times. Residence time over land is excluded from the integration of weighted trajectories. For satellite and modelled biological
variables, an 8-day average is necessary to obtain sufficient measurement spatial coverage (Section 2.2). While not ideal, an
8-day average is still useful because the phytoplankton cycle is fairly slow (1 year) relative to the frequency of meteorological
disturbances (days). In addition, advection is slower in the ocean as ocean currents are significantly slower than atmospheric
wind speed.

Weighted FLEXPART back trajectories are compared to particle properties and atmospheric DMS concentrations
measured aboard the *R/V Atlantis*. We define correlation strength by the calculated Pearson's coefficient (r) following Devore
and Berk (2012), where |r| < 0.25 indicates there is no correlation, $0.25 \leq |r| < 0.50$ is defined as a weak correlation, $0.50 \leq |r|$
<0.80 is defined as a moderate correlation, and $|r| \geq 0.80$ is defined as a strong correlation. For consistency, we have assumed
that the relationships between variables will be linear. Correlations of trajectories weighted by biological and meteorological
parameters with measured aerosol mass, number and DMS concentrations vary by the length of the weighted trajectory (Figure
4, Table S8-S16). In particular, many of these correlations increase with increasing trajectory length, indicating biological and
meteorological processes over the air mass history are influential to measured particle properties and that using only spatially
contemporaneous data would miss correlative connections that are lagged in time. When comparing measured quantities to 0-
5 day FLEXPART-weighted-residence-time explanatory variables, the slope of the linear regression generally flattens (or
decreases) with longer trajectories (Figure S5). This is because the trajectories cover more ocean surface area and thus they
are more likely to be weighted by both high and low values (for example, chlorophyll-a in Figure 3b). However, there are still
cases when a long (5-day) trajectory is consistently weighted by low values or high values, which may lead to extremes in
particle concentrations or composition, depending on the effect of the explanatory variable. In the following sections, the
correlation strength is interpreted based on known sources and sinks of marine particles and DMS.

### 3.1 Biological Controls on Marine Aerosols

Measured non-refractory organic aerosol mass correlates weakly with FLEXPART-residence-time-weighted chlorophyll-a,
with the highest correlation at 2-day trajectory lengths (r = 0.39; Figure 4a, 5a). The correlation of organic aerosol mass with
chlorophyll-a is similar for CDM (r = 0.32), but there is no correlation with sea water particulate organic carbon or the euphotic
zone depth (|r| < 0.25). Organic aerosol mass correlates moderately with trajectory weighted CAFÉ modelled net primary





production for 2 to 5-day trajectories (r = 0.54-0.62) (Figure 4a, 5b, Table S8). Comparisons of non-refractory organic aerosol mass with other net primary production models are shown in the supplemental Table S8. These results suggest a substantial portion of non-refractory organic mass is from secondary biogenic VOC emissions, such as isoprene and monoterpenes and other unidentified biogenic VOCs (Altieri et al., 2016; Hallquist et al., 2009). Non-refractory organic aerosol mass also

correlates with downward shortwave forcing (DSWF), with correlation strength increasing at longer trajectory lengths (Figure 4a, 5c). Increased solar radiation promotes photosynthesis by phytoplankton and is necessary for photochemical production of secondary organic aerosol. Biogenic VOC emissions (precursors of particle-phase organic mass) are a by-product of photosynthesis (Dani and Loreto, 2017), and likely cause part of the correlation of organic aerosol mass with the DSWF. Even though DSWF moderately correlates with organic aerosol mass concentrations (r = 0.52 at 2-day trajectories), the presence of

phytoplankton is necessary for photosynthetic biological emissions to occur (Silsbe et al., 2016). Figure 5c notably shows the DSWF is often higher during NAAMES4 (Mar.-Apr. 2018) compared to the other campaigns. However, NAAMES4 has lower measured organic aerosol mass concentrations, particularly when compared to the late spring campaign (NAAMES2, May-June 2016). The higher DSWF during much of NAAMES4 is simply due to the fact that the NAAMES4 campaign extended further to the south (to ~20°N, Figure 1d) than any of the other campaigns. The relatively low abundance of phytoplankton

biomass in the tropical Atlantic, compared to the subarctic Atlantic (shown by chlorophyll-a concentrations, Figure 1f,h,j,l), results in lower levels of photosynthesis, and therefore lower VOC emissions and organic aerosol mass formation. The amount of photosynthesis taking place depends on both the DSWF and phytoplankton abundance, both of which are included in the calculation of net primary production (in addition to other parameters, Section 2.3). Consequently, the phytoplankton net primary production is a better predictor of marine biogenic organic aerosol mass and likely biogenic VOC emissions, than

DSWF, chlorophyll-a biomass or any other individual biological parameter.

In contrast, the measured sulfate aerosol mass concentration has a weak or no correlation with the satellite measured biological parameters and modelled net primary production (r < 0.4), with the exception of the euphotic zone depth (r = 0.53 for 5-day trajectory), even though the main marine biogenic source of sulfate aerosol is from the oxidation of VOC emissions from phytoplankton, specifically DMS (Ayers and Gras, 1991; Bates et al., 1998a, 2012; Covert et al., 1992; Quinn et al., 2000;

Rinaldi et al., 2010; Sanchez et al., 2018). Measured atmospheric DMS concentrations moderately correlate with biological parameters for short trajectories (0-6 hours); however, this correlation is driven solely by a few measurements made during the bloom phase of the phytoplankton cycle (NAAMES2, Figure 6). Excluding the measurements from NAAMES2 would result in no significant correlation between measured atmospheric DMS and biological parameters. Significantly higher chlorophyll-a concentrations and net primary production rates are present during the NAAMES2 phytoplankton bloom,

causing the significantly higher DMS concentrations (Figure 1e-l). Unlike atmospheric DMS, DMS in seawater correlates moderately with the modelled net primary production even when excluding NAAMES2 (r = 0.54). This difference is driven by variability in the fraction of sea water DMS that is released into the atmosphere. In addition, DMS production is highly dependent on phytoplankton species (Keller, 1988), complicating the relationship between atmospheric DMS measurements and bulk ocean quantities and confounding direct correlations.



In addition, atmospheric DMS and non-refractory sulfate aerosol mass are observed to have little to no correlation with each other ($r = 0.34$) or with ocean biological activity. This lack of correlation may be due to the longer DMS atmospheric lifetime relative to biogenic VOCs (Kloster et al., 2006; Sciare et al., 2001). For a typical average OH radical concentration of 0.6 ppt, the lifetime of DMS is approximately 37 hours, which is much longer compared to the lifetime of known marine biogenic VOCs, such as isoprene (1.8 hours) and monoterpenes (10s of minutes to 3 hours depending on the species) (Atkinson and

Arey, 2003; Lee et al., 2006; Liakakou et al., 2007; Pandis et al., 1995; Seinfeld and Pandis, 2006). Organic aerosol mass likely correlates to sea surface biomass more than sulfate aerosol mass, partially because of the shorter lifetime. Once emitted, DMS can be advected large distances, or become lofted into the free troposphere (Clarke et al., 2013; Korhonen et al., 2008; Russell et al., 1998; Sanchez et al., 2018; Thornton et al., 1997), where the resulting sulfate aerosol is not sampled by the ship. For this reason, biomass and net primary production is not a good predictor of DMS (Gunson et al., 2006). Organic aerosol mass

precursors quickly condense onto existing particles to form SOA soon after emission (Ehn et al., 2014; Liakakou et al., 2007; Wennberg et al., 2018).

Similar to organic aerosol mass concentrations, particle number concentrations have weak correlations with biomass abundance (Figure 4d,e). Comparisons to particle number based on size are useful because they provide information on possible links to specific processes. The size distributions are split to derive integrated particle concentrations for diameters

greater and less than 100 nm ($N_{>100nm}$; $N_{<100nm}$; a rough delimiter between the Aitken- and accumulation-size modes) because of the differences in marine processes that contribute to changes in particle concentrations at different sizes. Specifically, Aitken-mode particle concentrations ($N_{<100nm}$) are strongly driven by new particle formation, which often occurs when total particle surface area is low (Humphries et al., 2015; Raes et al., 1997). In contrast, accumulation-mode particles concentrations ($N_{>100nm}$) are driven by the growth of the Aitken-mode, through cloud processing and condensation processes. $N_{>100nm}$ weakly

correlate with modelled net primary production ($r = 0.46$ for 5-day weighted back trajectories) and moderately correlate with non-refractory organic masses ($r = 0.65$) and strongly correlate with sulfate masses ($r = 0.83$); this is expected since larger particles account for most of the total submicron particle mass measured by the AMS. Small particles contain much less mass, so $N_{<100nm}$ have little to no correlation with the measured non-refractory organic ($r = 0.21$) and sulfate ($r = 0.39$) mass. $N_{<100nm}$ also has no correlations with chlorophyll-a ($r < 0.25$). This result does not preclude a biological influence on $N_{<100nm}$, but

demonstrates the limitations of this analysis in resolving some complex processes such as new particle formation. In addition, PMA concentrations correlate weakly with biomass abundance (Figure 4f), indicating there may be a weak influence of phytoplankton activity on PMA number concentration, as shown previously (Saliba et al., 2019). Weak correlations between particle number or composition with net primary production suggest other processes also influence particle concentrations. Possible meteorological influences are discussed in the following section.

**3.2 Meteorological Controls on Marine Aerosols**

Meteorological parameters can also influence particle concentrations. In addition to being important for biological processes, the DSWF is also important for initiating photochemical oxidation of biogenic VOCs, which leads to the formation of organic



and sulfate aerosol mass (Altieri et al., 2016; Bates et al., 1998b; Frossard et al., 2014b; Hallquist et al., 2009; O'Dowd et al., 2002; Rinaldi et al., 2010; Sievering et al., 1992a; Warren and Seinfeld, 1985). This process may cause the moderate correlation

between sulfate aerosol mass and DSWF despite having no correlation with satellite biomass or modelled net primary production (Figure 4, 5f). This also implies that DSWF is involved in both the photochemical oxidation of VOCs and photosynthesis of marine phytoplankton, and both processes likely strengthen the correlation of organic aerosol mass and DSWF. Figure 4 also shows how other meteorological parameters influence particle concentration. For example, low-level cloud cover is shown to negatively correlate with sulfate aerosol mass and $N_{>100nm}$. The presence of clouds will decrease the

solar radiation reaching the ocean surface, which would contributes to a negative correlation between low-level cloud cover and sulfate and $N_{>100nm}$, while aqueous production pathways would presumably result in a positive low-level cloud cover correlation with sulfate mass (Hoppel et al., 1986; Hudson et al., 2015; Pirjola et al., 2004; Sanchez et al. in prep, 2020). The negative correlation between low-level cloud cover and sulfate mass suggests the aqueous processing may be relatively less important than gas-phase photochemical mechanisms. Meteorological systems have consistent patterns, so, like low-level

cloud cover and DSWF, many other meteorological parameters covary with each other, complicating linking the particle properties to specific meteorological processes. Table S5 indicates DSWF correlates moderately or weakly with all the meteorological parameters considered from the GDAS data set. In this section, we examine how the other meteorological parameters, shown in Figure 4, also affect aerosol mass and number concentrations.

Sea surface wind speed negatively correlates with aerosol number and mass concentrations (Figure 4), except for PMA sized

particle number concentrations, and the correlation is increasingly strong for longer trajectory lengths. In order to understand the link between particle concentration and wind speed, it is important to recognize how wind speed may affect different particle formation processes (such as sea spray, particle growth, and particle formation). In contrast to submicron particle mass and number concentrations, sea surface wind speed positively correlates with PMA number concentrations (Figure 4f, 8a) (Saliba et al., 2019). This correlation is highest (r = 0.59) when only considering the wind speed at the ship location (0-hour

trajectories) and continuously decreases for longer back trajectory lengths. This indicates the shorter lifetime of PMA, due to the preferential sedimentation loss of larger particles, makes the history of the air mass irrelevant in this situation. The average dry PMA mode diameter, determined with a coarse-mode fitting algorithm, was 0.54 µm and as high as 1.12 µm (Saliba et al., 2019), which is larger than Aitken- and accumulation-mode particles. Because PMA particles are large, they are more prone to rapid removal from the MBL through deposition relative to smaller particles (Pryor and Barthelmie, 2000). Specifically, at

wind speeds greater than 10 m s$^{-1}$ the deposition dependent lifetime of a 3 µm wet-diameter particle in a 500 m MBL is about 3-12 hours, where the lifetime of a 0.1-1 µm wet-diameter particle is several days to weeks.

While counterintuitive, the inverse correlation between $N_{>100nm}$ and sea surface wind speed is likely driven by enhanced PMA concentrations at higher wind speeds that increase the pre-existing condensational sink for gas-to-particle conversion. PMA do not significantly contribute to total particle number concentrations (or $N_{>100nm}$), and incidentally are often considered not to

be major drivers of variability in cloud microphysics (Quinn et al., 2017). In a recent modelling study by Fossum et al. (2020), the authors showed evidence that the presence of elevated sea salt concentrations from PMA can indirectly affect cloud droplet





concentrations, by enhancing the uptake of water vapor at low supersaturations causing a reduction of in-cloud maximum supersaturations. A smaller maximum supersaturation will lead to fewer small particles ($N_{<100\ nm}$) activating to form cloud droplets and hence, fewer small particles growing through cloud processing to form large particles ($N_{>100nm}$). This feedback

does not explain why non-refractory sulfate and organic aerosol mass are also moderately and weakly inversely proportional to wind speed, respectively, but they are possibly also linked to PMA concentrations. PMA concentrations are fairly low compared to $N_{>100nm}$ and $N_{<100nm}$ (Figure 7c, d, 8a), but since PMA are quite large, they account for a significant fraction of the total particle surface area. Figure 8b shows the total particle surface area moderately correlates with 6-hour back trajectories weighted by sea surface wind speed (r = 0.51) and PMA frequently account for a majority of the total particle surface area.

Since new particle formation occurs under conditions of low total particle surface area, the enhancement in total particle surface area at elevated sea surface wind speeds can prevent the occurrence of new particle formation (Cainey and Harvey, 2002; Yoon and Brimblecombe, 2002). However, new particle formation is more likely to occur in the free troposphere, independently of MBL PMA concentration, then subsequently entrained into the MBL, possibly explaining the lack of correlation of $N_{<100nm}$ with wind speed (|r| = <0.25) (Clarke, 1993; Raes et al., 1997; Sanchez et al., 2018). The negative

correlation between non-refractory sulfate and organic mass with wind speed may be linked to the short lifetime of coarse PMA, relative to smaller particles. Consequently, any secondary particle formation through the condensation of VOC oxidation products and sulfate onto PMA are likely quickly removed through deposition. Up to 25% of secondary sulfate formation has been shown to form from aqueous ozone oxidation of $SO_2$ to sulfate on PMA particles (Sievering et al., 1992b). In addition, some of the secondary organic and sulfate mass may be missed by the AMS measurements. The AMS is limited to measuring

only non-refractory particles, so organic and sulfate aerosol mass that has condensed onto refractory PMA is less likely to be efficiently vaporized and measured (DeCarlo et al., 2006; Frossard et al., 2014a). Wind speed also inversely correlates with DSWF (r = -0.65, Table S5). As previously mentioned, DSWF is proportional to aerosol mass due to its stimulation of VOC emissions by marine biota and role in the photochemical oxidation of VOCs. Since meteorological variables covary, DSWF may partially drive the correlation strength between wind speed and aerosol mass concentrations, or vice versa.

Precipitation is a well-known sink for aerosol number and mass concentrations (Croft et al., 2010; Stevens and Feingold, 2009). In addition, observations of new particle formation events have been shown to correspond with precipitation during the air mass history, likely due to a decrease in large particle concentration and therefore total particle surface area because of precipitation scavenging (Andronache, 2004; Ueda et al., 2016). Despite this fact, precipitation was shown to have little to no correlation with aerosol number or mass (Figure 4, S4, r < 0.40). This observation may partially reflect the precipitation data being a 6-hour average GDAS product rather than direct measurements. The simulated precipitation estimates have been shown

to only moderately correlate with measurements (Beck et al., 2019). Figure 7c and 7d show the relationship between sea surface wind speed and $N_{>100nm}$ and $N_{<100nm}$. In this figure, the points are colored based on the log of the 5-day trajectory average precipitation and they show that, in general, elevated precipitation amounts typically correspond to higher wind speeds. When compared directly, wind speed and precipitation are still only weakly correlated (r = 0.38, Figure S4), but this weak

covariance indicates that the proposed links between elevated wind speeds and aerosol number and mass concentrations may



also be partially driven by enhanced precipitation. Precipitation will likely also remove the PMA; however, as shown in Figure 4f, PMA concentration has the highest correlation with recent wind speed and decreases with longer back trajectories. Even though PMA may be removed by precipitation, they are also replenished quickly.

## 4 Conclusions

We studied the relationship between marine aerosols measured over the North Atlantic Ocean during NAAMES with back trajectories weighted by four metrics of satellite measured ocean surface biomass (chlorophyll-a, sea water particulate organic carbon, colored detrital organic materials, euphotic zone depth), modelled net primary production, and model reanalysis meteorological parameters. Correlations between residence-time-weighted explanatory variables and aerosol measurements indicate both biological and meteorological processes influence the aerosol concentrations and composition. Specifically, non-

refractory organic aerosol mass correlates weakly with chlorophyll-a concentration, a proxy for phytoplankton abundance, averaged over 2-day back trajectories ($r = 0.39$) and moderately with net primary production over 5-day trajectories ($r = 0.62$). In general, the satellite derived chlorophyll-a, absorption coefficient of colored detrital organic materials, sea water particulate organic carbon and the euphotic zone depth moderately or strongly correlate with each other and therefore, had similar relationships to observed particle properties. In addition, organic aerosol mass moderately correlated with DSWF increasingly

for longer trajectory lengths ($r = 0.58$ at 5-day trajectories). These results indicate organic aerosol mass is influenced by the VOC emissions encountered by the air mass, which are driven by biological activity. Sulfate aerosol mass only weakly correlates with marine surface biomass ($r < 0.50$), even though marine non-sea-salt sulfate aerosol mass is also derived from marine VOC emissions. This difference is attributed to the short lifetime of organic aerosol mass precursors, like isoprene and monoterpenes (minutes to hours), which were below detection limits during NAAMES, relative to sulfate aerosol mass

precursors such as DMS (1-2 days) (Kloster et al., 2006; Liakakou et al., 2007; Sciare et al., 2001). The longer lifetime of DMS can delay the formation of sulfate aerosol mass, making sulfate precursors more likely to advect through long-range transport if vertically lofted into the free troposphere, and re-entrained down into the MBL. MBL to free troposphere transport of DMS is not captured well by the FLEXPART model.

Wind speed also weakly to moderately inversely correlated with aerosol concentration and mass. This relationship may be

driven by the enhanced formation of coarse-mode primary marine aerosol (PMA) at higher wind speeds. Specifically, enhanced PMA concentrations can prevent sulfate particles from activating to cloud droplets and growing through cloud processing. This result is consistent with a modelling study which indicated the enhanced rate of water vapor condensation onto PMA resulted in decreased cloud supersaturations (Fossum et al., 2020). The concentration of particles greater than 100 nm in diameter negatively correlated with wind speed ($r = -0.42$), consistent with this hypothesis; however sulfate ($r = -0.56$) and

organic ($r = -0.37$) aerosol mass are also negatively correlated with surface wind speed. We attribute this to the condensation of sulfate and VOCs onto PMA. While PMA accounts for a low fraction of the particle number, they are of larger size compared to most particles and are shown to account for $41\pm23\%$ of the total particle surface area. Large PMA particles have short lifetimes and deposit quickly relative to submicron particles (Pryor and Barthelmie, 2000). This is supported by the moderate



correlation between wind speed and PMA concentrations at 0-hour trajectories (r = 0.59), which decreased for longer trajectory
lengths. Condensed sulfate and organics on PMA in the MBL would quickly be removed from the atmosphere. In addition,
the AMS likely missed most secondary aerosol mass that condensed onto PMA because of its low efficiency for measuring
refractory components.

We find that air mass residence time is important for relating aerosol organics to ocean chlorophyll-a, net primary production
and downward shortwave forcing with moderate correlations observed for the longer simulation ages. Similarly, sulfate aerosol
showed moderate correlations with downward shortwave forcing and was anti-correlated with wind speed. Atmospheric DMS
and PMA concentrations showed better correlations with shorter back trajectory lengths, reflecting their origin as primary
emissions. In sum, this work demonstrates the need to account for air mass history when apportioning marine aerosol sources.
While this study seeks to understand linkages between the ocean and atmosphere, we have deliberately excluded the influence
of most continental transport to the marine atmosphere. Future studies are needed to understand and quantify the contribution
of transported aerosols to the marine CCN budget and how those may impact (or even dominate) the relationships we have
identified in the remote North Atlantic.

**Author Contribution**

Conceptualization, Methodology, and Writing - Original Draft: **KJS and RHM**; Software: **KJS and BZ**; Formal Analysis,
Visualization: **KJS**; Supervision, Project administration, Funding acquisition: **RHM, MHB, and LMR**; Data Curation: **GS,**
**CC, SLL, PKQ, TSB, JP, TGB, ESS, MJB;** Writing - Review & Editing: **all authors**.

**Acknowledgments**

We thank the dedicated crew of the R/V Atlantis. Kevin J. Sanchez was funded by the NASA Postdoctoral Program. The
authors also would like to acknowledge Raghu Betha, Derek Price, Derek Coffman, and Lucia Upchurch for collecting and
reducing data. We thank Emmanuel Boss for his input on the manuscript. We thank Nils Haentjens for processing the in-line
measurements. This work was funded by NASA grant NNX15AE66G, NNX15AF30G, NNX15AF31G and NSF grant
NSFOCE-1537943. This is PMEL contribution number 5111. Bo Zhang and Hongyu Liu acknowledge the funding support
from the NAAMES mission. The NAAMES dataset is archived in the NASA Atmospheric Science Data Center (ASDC;
https://doi.org/10.5067/Suborbital/NAAMES/DATA001) and the SeaWiFS Bio-Optical Archive and Storage System
(SeaBASS; https://doi.org/10.5067/SeaBASS/NAAMES/DATA001).
Scripps measurements are available at https://library.ucsd.edu/dc/collection/bb34508432. Shipboard measurements are
archived at https://seabass.gsfc.nasa.gov/. GlobColour data (http://globcolour.info) used in this study has been developed,
validated, and distributed by ACRI-ST, France. GDAS data are available at ftp://arlftp.arlhq.noaa.gov/pub/archives/gdas1/.
Modelled net primary production data are available at http://sites.science.oregonstate.edu/ocean.productivity/custom.php.



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



**Figure 1. (a-d) The residence time fraction of all marine 5-day FLEXPART back trajectories column integrated and normalized by the total residence time, (e-h) average satellite chlorophyll-a and (i-l) average CAFE modelled net primary production for each NAAMES campaign. The gray line in panels a-d shows the *R/V Atlantis* cruise track, while the black points are initialization points for the back trajectory model runs that satisfy the clean marine filter criteria (Section 2.6).**


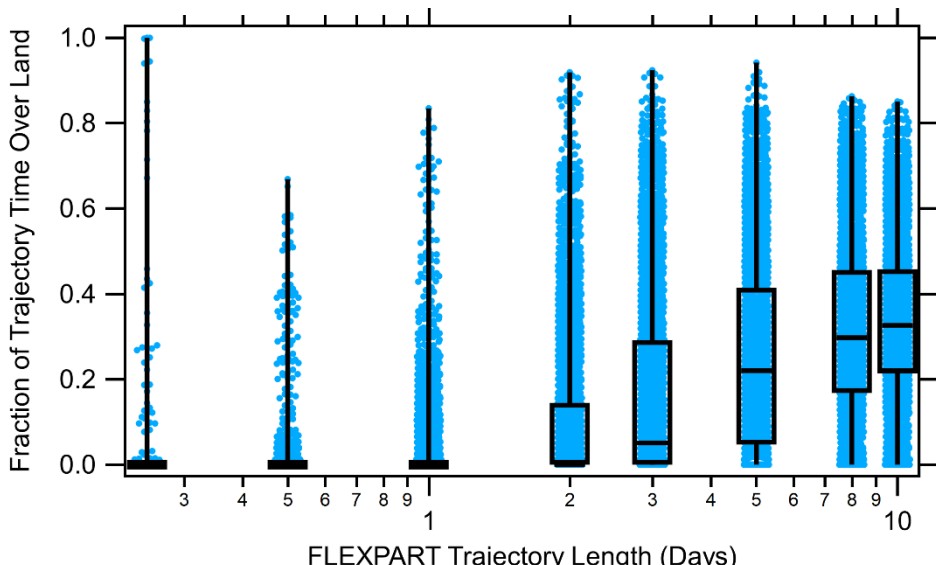

**Figure 2.** Boxplot of air mass residence time fraction spent over land for FLEXPART back trajectories of 6 hours to 10 days for all cases (both marine and continental back trajectories). Blue points represent each individual FLEXPART back trajectory. The horizontal lines show the 25th, 50th and 75th percentiles for the fraction of trajectory time over land and the vertical lines represent the range.






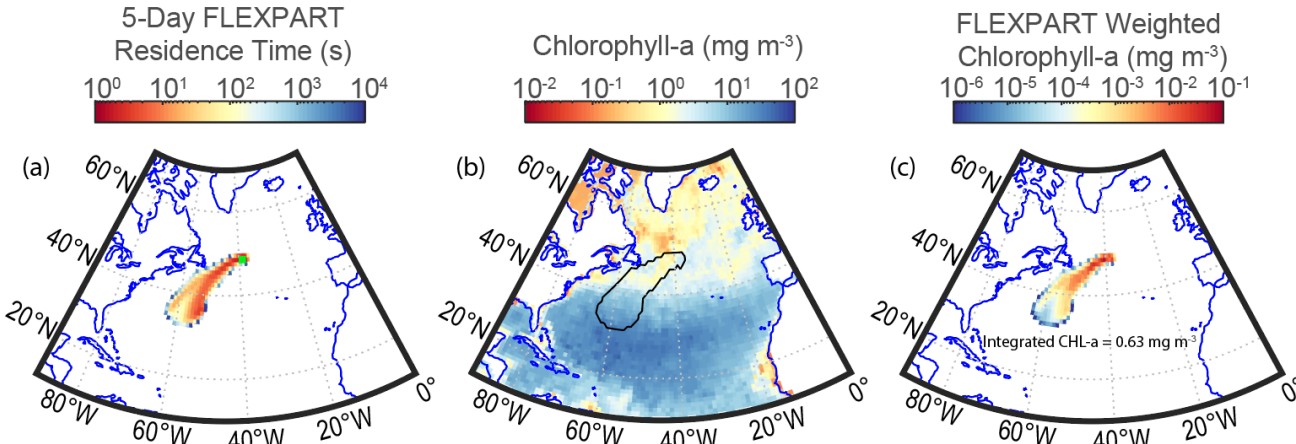

**Figure 3: (a) An example 5-day FLEXPART trajectory (MBL column-integrated) residence time distribution from NAAMES2 campaign initialized at 00z 19 May 2016 and the ship location, shown by the green square. (b) Satellite chlorophyll-a product and an outline of the 5-day FLEXPART back trajectory. (c) Satellite derived chlorophyll-a weighted by the 5-day FLEXPART residence time and then integrated to obtain a value representative of the level of influence chlorophyll-a may have had over the past 5 days on the air mass measured at the *R/V Atlantis* location.**





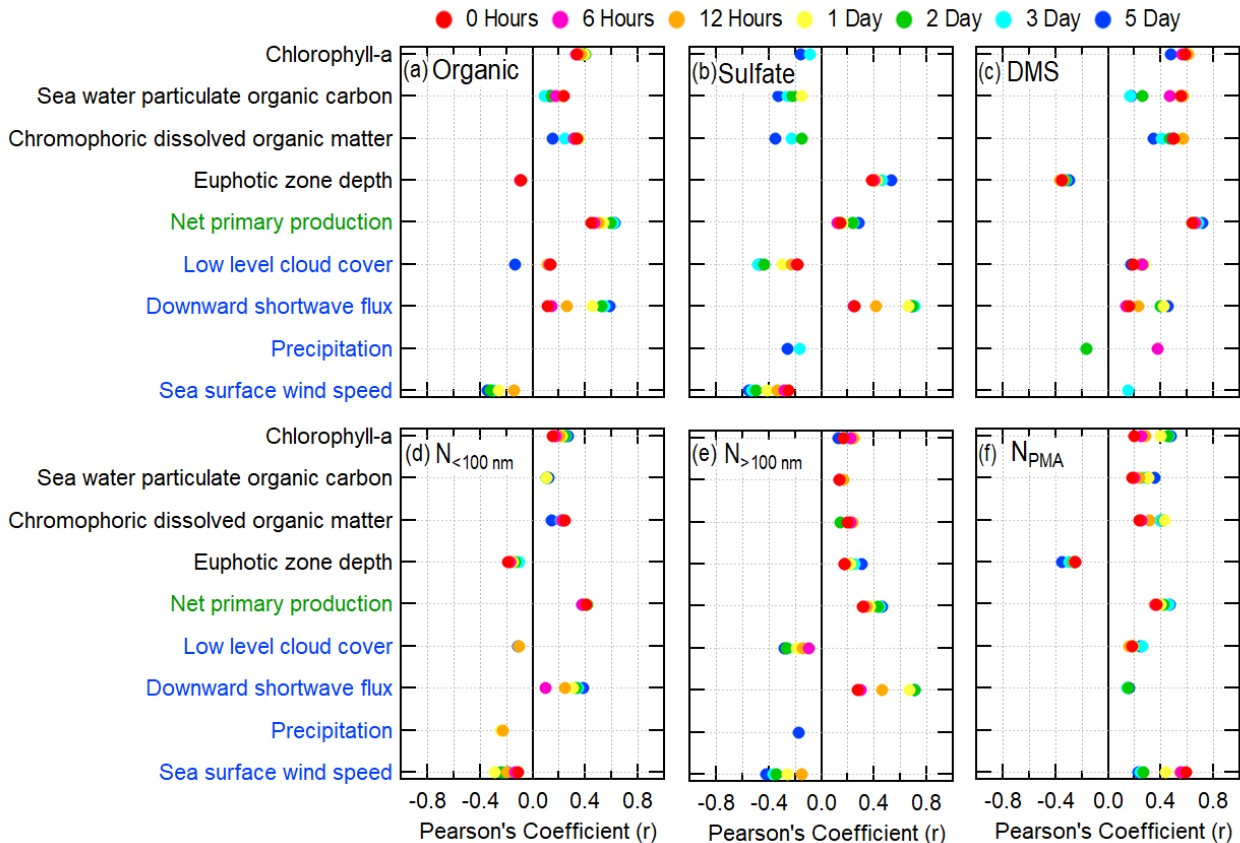

**Figure 4. Pearson's correlation coefficients between FLEXPART-residence-time-weighted explanatory variables and the following atmospheric measurement variables: (a) organic aerosol mass and (b) sulfate aerosol mass (c) DMS (d) particle number concentration for diameters < 100nm, (e) particle number concentration for diameters > 100nm, and (f) PMA mode concentration. The explanatory variables listed on the ordinate axis are colored to denote satellite-derived parameters (black text), CAFE ocean biology model parameters (green text), and atmospheric model reanalysis products (blue text). The PMA mode number concentration is derived from the SEMS and APS instruments on the *R/V Atlantis*. Pearson's correlation coefficients are only included for statistically significant cases where p < 0.05.**





**Figure 5. Measured atmospheric concentrations of (a-c) organic and (d-f) sulfate aerosol mass compared to 2-Day FLEXPART-residence-time-weighted satellite chlorophyll-a, modelled net primary production, and reanalysis model downward shortwave forcing. Pearson's coefficients (r) are included for each plot along with best fit lines shown as black lines.**








**Figure 6. Measurements of (a-c) atmospheric and (d-f) in-water DMS concentrations compared to *R/V Atlantis* measurements of chlorophyll, net primary production, and downward shortwave forcing. Pearson's coefficients (r) are included for each plot along with best fit lines shown as black lines.**




**Figure 7. Measurements of atmospheric (a) organic aerosol mass, (b) sulfate aerosol mass, (c) number concentration for diameters < 100 nm, and (d) number concentration for diameters > 100 nm compared to the 5-day FLEXPART-residence-time-weighted model reanalysis wind speed. Points are colored based on corresponding campaign in (a) and (b) and colored by the base 10 logarithm of the 5-day FLEXPART-residence-time-weighted model reanalysis 6-hour accumulated precipitation for (c) and (d). Pearson's coefficients (r) are included for each plot along with best-fit lines shown as black lines.**






**Figure 8. Measurements of (a) the PMA mode number concentration and (b) total particle surface area compared to 6-hour FLEXPART-residence-time-weighted model reanalysis wind speed. Both the total particle surface area and the PMA mode concentration are derived from the SEMS and APS instruments on the *R/V Atlantis*. Pearson's coefficients (r) are included for each plot along with best fit lines shown as black lines.**