# Peer review of "Linking marine phytoplankton emissions, meteorological processes and downwind particle properties with FLEXPART"

_Atmospheric Chemistry and Physics, 2020_

## Referee Comment (RC1) · Anonymous Referee #1 · 14 Oct 2020

General Comments

This manuscript presents the linking of marine aerosols to oceanic biological and meteorological parameters that were estimated by residence-time-weighted air mass transport history. The paper appears to be original and to provide a valuable dataset obtained during four field campaigns. However, there are a few scientific issues to be addressed before the paper can be accepted for publication.

Specific Comments

As the authors highlighted, the biological activities of the surface ocean have an important influence on the physiochemical properties of marine aerosols. Further clear

explanation of the biological characteristics of the study area would greatly benefit the paper. These biological characteristics may include the following: 'the main phytoplankton species, because the emission of biogenic VOCs is highly species-specific', 'differences in mean chlorophyll-a concentration and net primary production (NPP) for four field campaigns', and 'major biological pathway in oceanic VOCs production'.

Air mass transport history, combined with biological and meteorological parameters, was used to estimate environmental factors controlling marine aerosols and VOC in this study. Similar analyses have been conducted previously (Arnold et al., 2010; Park et al., 2018); hence, the authors should definitely explicitly explain what aspects of their work are novel and of significance.

The authors need to provide time series measurement results for key observation parameters (including atmospheric concentration of organics, sulfate and DMS).

Biogenic VOCs in the ocean can be produced via several pathways, including photosynthetic byproducts, bacterial degradation of dissolved organic matter, and zooplankton grazing on marine phytoplankton. The authors need to verify relevant explanations (e.g., lines 32-33).

Provide relevant references for lines 46-48 and lines 54-55.

Line 56: dissolved organic matter also acts as an important contributor to marine aerosols.

Line 66: In general, the abundance of marine phytoplankton reaches its maximum during the spring period, and the mixed layer depth is much shallower during summer than during spring.

Line 93: Chlorophyll-a could be used as an indicator for the biomass of marine phytoplankton, but not for biogenic VOC emissions. The production of biogenic VOCs is highly species-specific and is controlled by a complex food-web mechanism.

Lines 213-214: NPP is not the only process that is linked with biogenic VOC emissions;

oceanic VOC production is also related to multiple biological processes. Please modify this sentence.

Line 273: "because the phytoplankton cycle is fairly slow (1 year)". I agree that the use of 8-day averaged values for sea surface chlorophyll-a and NPP is sufficient to evaluate the relation between aerosol parameters and oceanic biological activities. However, this explanation is inadequate because the life cycle of individual phytoplankton is not that slow (typically a few days). I believe that it might be better to demonstrate the variation in daily (or 8-day) chlorophyll-a concentration at a given domain for each cruise period to support the idea that the use of 8-day chlorophyll-a values is appropriate.

Lines 284-286: Comparing in-line chlorophyll-a with trajectory-weighted chlorophyll-a does not make sense. This is because the FLEXPART backward trajectories reflect the travel history of air parcels rather than ocean currents.

As shown in Fig. 4, several key aerosol parameters are weakly and moderately correlated with FLEXPART-residence-time-weighted explanatory variables when all datasets obtained from the four separate field campaigns that were conducted in different seasons are gathered. However, to clearly support the author's explanation, a statistically valid relation between these parameters should be observed for each cruise. This is because Environmental factors affecting the formation and growth of marine aerosols may vary from season to season.

It would be better to provide figures for 5-day FLEXPART residence-time-weighted values, since the authors insist that many of the correlation strength increased at longer trajectory lengths.

Line 436: The use of 'satellite measured ocean surface biomass' is not correct. This is because colored detrital organic materials and euphotic zone depth do not reflect biomass at the sea surface.

Line 440: replace 'abundance' with 'biomass'

Line 447: What does 'surface biomass' indicate? 'Net primary production'? NPP does not mean biomass.

Provide a clear explanation for 'refractory' and 'non-refractory' particles.

Lines 31-35 (abstract), 445-453 (conclusion), and relevant explanation in the Results and Discussion section: The explanation in these parts is confusing and hard to follow. As the authors noted, the lifetime of DMS in the atmosphere (1-2 days) is longer than that of other trace gases such as isoprene and monoterpene (less than a few hours). Considering the typical growth rate of SOA particles in the marine atmosphere, the difference in the lifetime of these VOCs (DMS, isoprene, etc.) may not significantly affect their temporal contribution to the organic aerosol mass over the study period. Moreover, the North Atlantic Ocean is well-known for high sea water DMS concentrations (a few nM, and occasionally increasing up to hundreds of nM during the phytoplankton bloom period due to the high abundance of DMS-producing phytoplankton groups such as haptophytes). The seawater concentration of isoprene (a few pM) in the North Atlantic Ocean is much lower than that of DMS (e.g., Dani and Loreto, 2017).

Technical corrections

Line 68: replace SO4 with SO42-

Line 168: provide full name for SEMS

References

Arnold, S.R., Spracklen, D.V., Gebhardt, S., Custer, T., Williams, J., Peeken, I., Alvain, S., 2010. Relationships between atmospheric organic compounds and air-mass exposure to marine biology. Environ. Chem. 7 (3), 232–241. https://doi.org/10.1071/EN09144.

Park, K.-T., Lee, K., Kim, T.-W., Yoon, Y.J., Jang, E.-H., Jang, S., Lee, B.-Y., Hermansen, O., 2018. Atmospheric DMS in the Arctic Ocean and Its Relation to Phytoplankton Biomass. Global Biogeochem. Cy. 32 (3), 351–359.

https://doi.org/10.1002/2017GB005805.

Dani, K. G. S., and Loreto, F. (2017). Trade-off between dimethyl sulfide and iso-prene emissions from marine phytoplankton. Trends Plant Sci. 22, 361–372. doi: 10.1016/j.tplants.2017.01.006

---

## Referee Comment (RC2) · Anonymous Referee #2 · 15 Oct 2020

Review of the manuscript "Linking marine phytoplankton emissions, meteorological processes and downwind particle properties with FLEXPART" by Kevin J. Sanchez et al.

**General comment:**
The manuscript presents how observed aerosol particle composition and concentrations are correlated with different meteorological and biological processes in the ocean upwind of the measurement location. The analysis is performed using meteorological data, comprehensive aerosol observations and the FLEXPART Lagrangian particle dispersion model. I find most of the results convincing and well justified by known physical/biological mechanisms. The paper is also generally well written and easy to follow. I suggest that the paper should be accepted for publication in ACP after minor revision considering all reviewers comments.

**Specific comments:**

Abstract: L39-41 "We hypothesized that the elevated total particle surface area associated with high PMA concentrations leads to enhanced rates of VOC condensation onto PMA" It is not the VOCs but their low- or semi-volatile oxidation products that condenses onto the PMA. Please, modify this sentence.

The observed negative correlation between aerosol submicron number and mass concentrations and wind speeds can as the authors suggest partly be a result of enhanced rate of condensation of condensable vapors onto wind generated PMA. However, other factors such as coagulation sink, and enhanced vertical mixing and dry deposition losses during high wind speed conditions may also contribute to the observed correlation.

P8, L240-242 "For the remaining analysis in this paper, the vertical structure of the residence time is column integrated over only the vertical levels that are completely or partially within the MBL based on GDAS MBL heights. Remaining vertical levels were excluded from analysis."

Is it reasonably to exclude the air mass residence time above the MBL from the analysis? Does, this not also exclude the impact of free tropospheric air masses which may dilute the observed MBL aerosol concentrations. I would like to see result on how large fraction of the air mass residence time which is excluded because a fraction of the FLEXPART particles tracers is above the MBL. I would also like to see some analysis on if the fraction of air masses which is above the MBL correlates (anti-correlates) with the observed aerosol concentrations. I would expect that a large contribution from free troposphere air masses would result in lower PMA and aerosol particle mass in general, but possibly higher particle number concentrations.

P9, L271 "Residence time over land is excluded from the integration of weighted trajectories" Similar comment as above. Is it reasonable to exclude the residence time over continents. Should this residence time not be included in equation 1 but with the d explanatory variable values ($E_t$) set to zero or a value representing e.g. emissions over the continents?

P9, L277-279 "We define correlation strength by the calculated Pearson's coefficient (r) following Devore and Berk (2012), where $|r| < 0.25$ indicates there is no correlation, $0.25 \leq |r| < 0.50$ is defined as a weak correlation, $0.50 \leq |r| < 80$ is defined as a moderate correlation, and $|r| \geq 0.80$ is defined as a strong correlation."

What I miss in the main manuscript (at least I could not find it), but what is included in the supplementary tables, is a statement about if the correlation coefficient is significantly separated from zero (r=0). Please add a sentence stating e.g. that only statistically significant correlations on a 5 % significance level (p<0.05) is presented.

P10, L296-299 "Comparisons of non-refractory organic aerosol mass with other net primary production models are shown in the supplemental Table S8. These results suggest a substantial portion of non-refractory organic mass is from secondary biogenic VOC emissions, such as isoprene and monoterpenes and other unidentified biogenic VOCs (Altieri et al., 2016; Hallquist et al., 2009)."

What about MSA formed from DMS? In the AMS I expect that the MSA mass will be assigned both to the sulfate and organics non-refractory mass.

P12, L373-374 "The negative correlation between low-level cloud cover and sulfate mass suggests the aqueous processing may be relatively less important than gas-phase photochemical mechanisms."

I expect that low-level cloud cover also correlates with precipitation. Can this not also contribute to the negative correlation between the sulfate mass and low level clouds?

P13, L412-413 "Up to 25% of secondary sulfate formation has been shown to form from aqueous ozone oxidation of $SO_2$ to sulfate on PMA particles (Sievering et al., 1992b)"

Yes, this may be correct but generally the most important aqueous phase $SO_2$ oxidation mechanism leading to sulfate is the reaction between $H_2O_2$ and $SO_2$.

P14, L450-453 "The longer lifetime of DMS can delay the formation of sulfate aerosol mass, making sulfate precursors more likely to advect through long-range transport if vertically lofted into the free troposphere, and re-entrained down into the MBL. MBL to free troposphere transport of DMS is not captured well by the FLEXPART model."

I agree. What complicates things with DMS is that the DMS oxidation is a multiphase process involving both gas- and aqueous phase and OH, O3 and halogens. The fraction of DMS which is oxidized to $SO_2$ will delay the sulfate aerosol mass even further. In the gas-phase $SO_2$ has a relatively long lifetime (~1 week).

---

## Author Comment (AC1) · 13 Nov 2020

**General comment:**

The manuscript presents how observed aerosol particle composition and concentrations are correlated with different meteorological and biological processes in the ocean upwind of the measurement location. The analysis is performed using meteorological data, comprehensive aerosol observations and the FLEXPART Lagrangian particle dispersion model. I find most of the results convincing and well justified by known physical/biological mechanisms. The paper is also generally well written and easy to follow. I suggest that the paper should be accepted for publication in ACP after minor revision considering all reviewers comments.

**Specific comments:**

Abstract: L39-41 "We hypothesized that the elevated total particle surface area associated with high PMA concentrations leads to enhanced rates of VOC condensation onto PMA" It is not the VOCs but their low- or semi-volatile oxidation products that condenses onto the PMA. Please, modify this sentence.
We have updated the sentence to state: "…enhanced rates of condensation of VOC oxidation products onto PMA."

The observed negative correlation between aerosol submicron number and mass concentrations and wind speeds can as the authors suggest partly be a result of enhanced rate of condensation of condensable vapors onto wind generated PMA. However, other factors such as coagulation sink, and enhanced vertical mixing and dry deposition losses during high wind speed conditions may also contribute to the observed correlation.

We agree and have added the following sentence to section the section 3.2:
    "Other factors associated with high wind speed conditions, such as coagulation, dry deposition, and enhanced vertical mixing likely also contribute to the observed negative correlation between wind speed and particle concentration."

P8, L240-242 "For the remaining analysis in this paper, the vertical structure of the residence time is column integrated over only the vertical levels that are completely or partially within the MBL based on GDAS MBL heights. Remaining vertical levels were excluded from analysis."
Is it reasonably to exclude the air mass residence time above the MBL from the analysis? Does, this not also exclude the impact of free tropospheric air masses which may dilute the observed MBL aerosol concentrations. I would like to see result on how large fraction of the air mass residence time which is excluded because a fraction of the FLEXPART particles tracers is above the MBL. I would also like to see some analysis on if the fraction of air masses which is above the MBL correlates (anti-correlates) with the observed aerosol concentrations. I would expect that a large contribution from free troposphere air masses would result in lower PMA and aerosol particle mass in general, but possibly higher particle number concentrations.

We have performed the analysis while including the residence time above the MBL and the results were not significantly different (See figure below after next comment). This shows that most air masses were largely confined in the boundary layer up to 5 days because of the stable nature of the MBL. Though mixing of FT air into the MBL can happen, the amount and occurrence is low enough that their influence cannot be seen in the statistics. Also, like all global

models, the GFS's coarse resolution prevents it from resolving some of the small scale mixing for FT and MBL air, but is well suited for resolving the large-scale advection. Furthermore, since we are examining how the ocean surface and boundary layer meteorology influence marine boundary layer particles, we do not believe it is reasonable to weight FT residence time by boundary layer quantities.

P9, L271 "Residence time over land is excluded from the integration of weighted trajectories" Similar comment as above. Is it reasonable to exclude the residence time over continents. Should this residence time not be included in equation 1 but with the d explanatory variable values ($E_t$) set to zero or a value representing e.g. emissions over the continents?

We agree that including the continental residence time and setting the explanatory variable values to zero is an alternative approach. However, this approach is equivalent to stating that there is zero influence from continental processes and, for this reason, we believe our approach is better. Despite this choice, we have performed the suggested approach of including the continental residence time (and included FT residence time) and the results are not significantly different (see figure below). Furthermore, extensive filtering for continentally influenced air (and specifically high continental residence time) was performed to remove cases that were significantly influenced by continental processes.

[Figure]

Reviewer figure 1: Same as figure 4 of main text, but FLEXPART trajectories that extend into the FT and over land are included (explanatory marine biological variables = 0 over land).

P9, L277-279 "We define correlation strength by the calculated Pearson's coefficient (r) following Devore and Berk (2012), where $|r| < 0.25$ indicates there is no correlation, $0.25 \leq |r| < 0.50$ is defined as a weak correlation, $0.50 \leq |r| < 80$ is defined as a moderate correlation, and $|r| \geq 0.80$ is defined as a strong correlation."
What I miss in the main manuscript (at least I could not find it), but what is included in the supplementary tables, is a statement about if the correlation coefficient is significantly separated from zero (r=0). Please add a sentence stating e.g. that only statistically significant correlations on a 5 % significance level ($p<0.05$) is presented.

We agree and have added the following sentence:
    "All presented Pearson's correlation coefficients are statistically significant ($p < 0.05$)."

P10, L296-299 "Comparisons of non-refractory organic aerosol mass with other net primary production models are shown in the supplemental Table S8. These results suggest a substantial

portion of non-refractory organic mass is from secondary biogenic VOC emissions, such as isoprene and monoterpenes and other unidentified biogenic VOCs (Altieri et al., 2016; Hallquist et al., 2009).”

What about MSA formed from DMS? In the AMS I expect that the MSA mass will be assigned both to the sulfate and organics non-refractory mass.

Yes, MSA will contribute to both organic and sulfate non-refractory mass for AMS unit mass resolution measurements. We have updated the sentence to also include MSA as a contributing VOC to the overall organic mass and added a relevant reference (Zorn et al., 2008).

P12, L373-374 “The negative correlation between low-level cloud cover and sulfate mass suggests the aqueous processing may be relatively less important than gas-phase photochemical mechanisms.”

I expect that low-level cloud cover also correlates with precipitation. Can this not also contribute to the negative correlation between the sulfate mass and low level clouds?

We have looked into the influence of precipitation on aerosol properties (see figure 4) and found no significant correlation with precipitation.  However, as mentioned in the manuscript, the reanalysis precipitation has been shown to only moderately correlate with observations (Beck et al., 2019) and we believe the role of precipitation is possibly more important than what our analysis shows.

For reference, below is a figure showing the relation between the 2-Day trajectories averaged precipitation and low cloud fraction.

[Figure]

Reviewer figure 2: 2-Day trajectory weighted Precipitation and low cloud fraction from the GDAS reanalysis.

P13, L412-413 “Up to 25% of secondary sulfate formation has been shown to form from aqueous ozone oxidation of SO2 to sulfate on PMA particles (Sievering et al., 1992b)”

Yes, this may be correct but generally the most important aqueous phase SO2 oxidation mechanism leading to sulfate is the reaction between H2O2 and SO2.

Yes under most conditions, H2O2 is the dominate oxidant for aqueous phase sulfate formation from SO2. We have updated this statement to take into account this reaction:

> "Secondary sulfate formation from $SO_2$ occurs rapidly on fresh PMA particles via uptake due to aqueous ozone reactions (Sievering et al., 1992). Subsequently, in-cloud sulfate formation from $SO_2$ continues by oxidation due to hydrogen peroxide (Jacob, 2000)."

P14, L450-453 "The longer lifetime of DMS can delay the formation of sulfate aerosol mass, making sulfate precursors more likely to advect through long-range transport if vertically lofted into the free troposphere, and re-entrained down into the MBL. MBL to free troposphere transport of DMS is not captured well by the FLEXPART model."
I agree. What complicates things with DMS is that the DMS oxidation is a multiphase process involving both gas- and aqueous phase and OH, O3 and halogens. The fraction of DMS which is oxidized to SO2 will delay the sulfate aerosol mass even further. In the gas-phase SO2 has a relatively long lifetime (~1 week).
We agree and have added the following text to the results section:

> "DMS also has a number of chemical pathways with various secondary aerosol yields, making a direct link to biological processes more challenging (Faloona et al., 2009) "

And

> "Also, $SO_2$, a DMS oxidation product, has a lifetime on the order of days to weeks."

We have also updated the below text in the conclusion:

> "Furthermore, the longer lifetime of DMS and its oxidation products can delay the formation of sulfate aerosol mass, making sulfate precursors more likely to advect through long-range transport if vertically lofted into the free troposphere and re-entrained down into the MBL. MBL to free troposphere transport of DMS is not captured well by the FLEXPART model. In addition, there are numerous DMS chemical pathways with various secondary aerosol yields that can obscure any link between sulfate aerosol concentrations and biogenic processes (Faloona et al., 2009)."

---

## Author Comment (AC2) · 13 Nov 2020

**General Comments**

This manuscript presents the linking of marine aerosols to oceanic biological and meteorological parameters that were estimated by residence-time-weighted air mass transport history. The paper appears to be original and to provide a valuable dataset obtained during four field campaigns. However, there are a few scientific issues to be addressed before the paper can be accepted for publication.

We sincerely appreciate the reviewer's feedback and believe their comments have led to an improved manuscript. We have responded to the reviewer's comments with the blue text below.

**Specific Comments**

As the authors highlighted, the biological activities of the surface ocean have an important influence on the physiochemical properties of marine aerosols. Further clear explanation of the biological characteristics of the study area would greatly benefit the paper. These biological characteristics may include the following: 'the main phytoplankton species, because the emission of biogenic VOCs is highly species-specific', 'differences in mean chlorophyll-a concentration and net primary production (NPP) for four field campaigns', and 'major biological pathway in oceanic VOCs production'.

We agree and have added the additional details that the reviewer is seeking to the Introduction, Methods, and Conclusions as follows:

In Introduction:

"Analysis of phytoplankton taxonomy and its seasonal variability in the NAAMES region is presented by Bolanos et al. (2020). Bolanos et al. (2020) show cyanobacteria dominated subpolar waters during the winter and were a significant fraction in the subtropics, with taxa varying by latitude. In-addition, prasinophyta accounted for a significant contribution of subtropical species, with stramenopiles representing less than 30% of subtropical communities. Spring communities had significantly more diverse communities and significantly less cyanobacteria (<10%) relative to the winter, with the exception of one station. Prasinophyta dominated the spring phytoplankton composition, though taxonomic compositions differed from the winter period and between the subpolar and subtropical regions. Typically, diatoms are assumed to be the dominant phytoplankton species in blooms. However, diatoms only represent 10-40% of phytoplankton biomass in the spring bloom surveyed during NAAMEs. The phytoplankton functional groups present influence the overall isoprene production rate and, therefore, the marine atmospheric aerosol and VOC concentrations. Booge et al. (2016) compiled chlorophyll-normalized isoprene production rates from the literature to identify differences between phytoplankton species. The chlorophyll-normalized isoprene production rates varied from 4.56-27.6, 1.4-32.16, and 1.12-28.48 ($\mu$mol (g chlorophyll-a) $^{-1}$ day$^{-1}$) for cyanobacteria, prasinophyta and diatoms, respectively, indicating emissions for isoprene vary significantly with taxonomy. To further complicate the emission strength of VOCs, emissions can vary by production pathways, such as

photosynthetic byproducts, bacterial degradation of dissolved organic matter, and zooplankton grazing on marine phytoplankton (Gantt et al., 2009; Shaw et al., 2003; Sinha et al., 2006)."

In methods:
"For this study chlorophyll-normalized VOC production rates were not considered because of the large variability in observed values (Booge et al., 2016) and the overall unknown contributions from various VOC species to marine particle mass concentrations."

Updated text in conclusion:

"Future studies are needed to 1) understand how differences in subtropical and subarctic phytoplankton speciation may influence aerosol concentrations (Bolaños et al., 2020) and 2) quantify the contribution of transported aerosols to the marine CCN budget and how those may impact (or even dominate) the relationships we have identified in the remote North Atlantic."

Air mass transport history, combined with biological and meteorological parameters, was used to estimate environmental factors controlling marine aerosols and VOC in this study. Similar analyses have been conducted previously (Arnold et al., 2010; Park et al., 2018); hence, the authors should definitely explicitly explain what aspects of their work are novel and of significance.

We thank the reviewer for pointing out this literature. The analysis performed by Arnold et al. (2010) and Park et al. (2018) are similar, however what really sets our manuscript apart is that we compared the influence of biomass on aerosol rather than VOCs (with the exception of DMS). We have added the references in the following relevant sentence in the introduction:
"Previous literature hints that phytoplankton activity is related to emissions of organic and sulfate aerosol mass precursors (Altieri et al., 2016; Arnold et al., 2010; Ayers et al., 1997; Bates et al., 1998; Brüggemann et al., 2018; Ceburnis et al., 2011; Facchini et al., 2008; Hallquist et al., 2009; Hu et al., 2013; Huang et al., 2018; Mansour et al., 2020; Ovadnevaite et al., 2014; Park et al., 2017; Quinn et al., 2019; Sanchez et al., 2018)."

We have also added the following statement in the results section:
"The DMS correlation with chlorophyll-a during the bloom period is consistent with results from similar analyses performed by Arnold et al. (2010) and Park et al. (Park et al., 2018), where DMS measurements were collected in the South Atlantic and Arctic, respectively."

The authors need to provide time series measurement results for key observation parameters (including atmospheric concentration of organics, sulfate and DMS).
The following figure has been added to the supplement and referenced in the main text:

[Figure]

Figure S1. Time series of hourly CN and CN$_{>100nm}$, non-refractory organic and sulfate concentration, DMS concentration and *R/V Atlantis* latitude for each NAAMES campaign. Data has been filtered for clean marine conditions (see section 2.6).

Biogenic VOCs in the ocean can be produced via several pathways, including photosynthetic byproducts, bacterial degradation of dissolved organic matter, and zooplankton grazing on marine phytoplankton. The authors need to verify relevant explanations (e.g., lines 32-33).

We agree that we cannot link emissions to photosynthetic byproducts alone and have updated the text:

Updated text:
> "This result indicates non-refractory organic aerosol mass is influenced by biogenic volatile organic compound (VOC) emissions that are typically produced through bacterial degradation of dissolved organic matter, zooplankton grazing on marine phytoplankton, and as a byproduct of photosynthesis by phytoplankton stocks during advection into the region."

Provide relevant references for lines 46-48 and lines 54-55.
We have added relevant references for lines 46-48:
> "Marine environments are sensitive to aerosol particle loading because particles can act as cloud condensation nuclei (CCN) on which cloud droplets form. The number concentration of cloud droplets can influence cloud optical properties and therefore affect the impact of clouds on climate (Leahy et al., 2012; Platnick and Twomey, 1994; Turner et al., 2007; Warren et al., 1988)."

Lines 54-55 are simply a statement. We have reworded the statement to prevent confusion:
> "Since ocean-emitted volatile compounds and particles can control the number, size and composition of marine aerosols (Brooks and Thornton, 2018), here we use satellite measurements of ocean biomass as a proxy for marine particle properties."

Line 56: dissolved organic matter also acts as an important contributor to marine aerosols.

We have updated the text to included DOM as a contributor to marine aerosols.

Line 66: In general, the abundance of marine phytoplankton reaches its maximum during the spring period, and the mixed layer depth is much shallower during summer than during spring.

We have updated the text below:
> "The bloom ends when phytoplankton division rates stop increasing (due to depletion of nutrients or annual maximum in mixed layer light intensity) and are matched by loss rates (Behrenfeld and Boss, 2018). When bloom termination is associated with nutrient exhaustion, mixed later depths may continue to shoal into summer (i.e., mixed layer light levels are still increasing), but phytoplankton biomass may decrease due to slowing division rates and excessive grazing (Behrenfeld and Boss, 2018)."

Line 93: Chlorophyll-a could be used as an indicator for the biomass of marine phytoplankton, but not for biogenic VOC emissions. The production of biogenic VOCs is highly species-specific and is controlled by a complex food-web mechanism.

We agree and have changed "biogenic VOC emissions" to "marine phytoplankton biomass".

Lines 213-214: NPP is not the only process that is linked with biogenic VOC emissions; oceanic VOC production is also related to multiple biological processes. Please modify this sentence.
The text has been updated as below:
> "Net primary production is the formation of organic material through photosynthesis by phytoplankton. This process and correlated changes in other ecosystem rates lead to the emission of biogenic VOCs at the sea surface (Li et al., 2018). "

While NPP is not the only processes that produces biogenic VOC emissions, we are directly comparing NPP to aerosol and believe it is correct to state that any correlation is most likely related to VOCs that are produced as a byproduct of NPP and not other processes.
We have updated text elsewhere (as pointed out by the reviewer in other comments) to properly state the other sources of biogenic VOC emissions in more general statements.

Line 273: "because the phytoplankton cycle is fairly slow (1 year)". I agree that the use of 8-day averaged values for sea surface chlorophyll-a and NPP is sufficient to evaluate the relation between aerosol parameters and oceanic biological activities. However, this explanation is inadequate because the life cycle of individual phytoplankton is not that slow (typically a few days). I believe that it might be better to demonstrate the variation in daily (or 8-day) chlorophyll-a concentration at a given domain for each cruise period to support the idea that the use of 8-day chlorophyll-a values is appropriate.
The reviewer is referring to the following text:
> "While not ideal, an 8-day average is still useful because the phytoplankton cycle is fairly slow (1 year) relative to the frequency of meteorological disturbances (days)"

We agree that demonstrating the lack of variation in consecutive 8-day average chlorophyll-a concentrations is an effective method to support the use of an 8-day average and have added the following text at the end of the sentence:
> "and consequently the low variation from one 8-day average in Chlorophyll-a values to the next indicates an 8-day average is appropriate (Figure S5)."

The new supplementary figure is shown below:

[Figure]

Figure S5. The normalized distribution of the difference in Chlorophyll-a between two consecutive satellite 8-day averages (24 May 2016 – 1 Jun 2016, shown in Figure 3b, and 1 Jun 2016 and 9 Jun 2016). The distribution includes the difference in chlorophyll-a from every 1° x 1° cell between 0° W and 90° W, and 10° N and 70° N, excluding cells on continents or with missing values.

Lines 284-286: Comparing in-line chlorophyll-a with trajectory-weighted chlorophyll-a does not make sense. This is because the FLEXPART backward trajectories reflect the travel history of air parcels rather than ocean currents.

The reviewer is referring to the initial Figure S5 referenced in the following text:

> "When comparing measured quantities to 0-5 day FLEXPART-weighted-residence-time explanatory variables, the slope of the linear regression generally flattens (or decreases) with longer trajectories (Figure S5). This is because the trajectories cover more ocean surface area and thus they are more likely to be weighted by both high and low values (for example, chlorophyll-a in Figure 3b)."

Our intentions were to highlight the fact that there is less dependence on the local chl-a concentration when considering longer trajectory lengths. After consideration, we have decided to remove the figure to prevent confusion. The text has been updated as follows:

> "Over longer trajectories, the weighted parameter is less likely to be related to the local value because the trajectories cover more ocean surface area and thus they are more likely to be weighted by both high and low values (for example, chlorophyll-a in Figure 3b)."

As shown in Fig. 4, several key aerosol parameters are weakly and moderately correlated with FLEXPART-residence-time-weighted explanatory variables when all datasets obtained from the four separate field campaigns that were conducted in different seasons are gathered. However, to clearly support the author's explanation, a statistically valid relation between these parameters

should be observed for each cruise. This is because Environmental factors affecting the formation and growth of marine aerosols may vary from season to season.

We agree that, to some unknown degree, the statistics are likely influenced by difference in environmental factors from one season to the next. However, Figure 4 is already quite a lot of information to process and separating this analysis by season would generate 4 times the information. Furthermore, the dynamic range of observed marine biological parameters would be significantly lower for individual seasons, likely resulting in statistically insignificant relationships between biological processes and aerosol properties. Separating by season would also significantly reduce the sample size. Such an analysis may be appropriate by combining a number of campaigns that occurred during the same season.

It would be better to provide figures for 5-day FLEXPART residence-time-weighted values, since the authors insist that many of the correlation strength increased at longer trajectory lengths.

In Figure 5, 2-day FLEXPART residence-time-weighted values were used because the 2-day trajectory corresponded to the peak in the correlation between non-refractory organic aerosol mass and Chlorophyll-a. While Chlorophyll-a does not have the strongest correlation, its importance lies in the fact that it is commonly used as a proxy for marine biomass and marine biogenic particle production. Also the peak correlation between organic aerosol mass and NPP and DSWF (Figure 5b,c,e,f) is similar to the 2-day trajectory correlation value. In Figure 6, DMS was shown to correlate with shorter trajectory lengths when comparing to Chlorophyll-a and net primary production, so it made more sense to compare to a low trajectory length (0-hour).

Line 436: The use of 'satellite measured ocean surface biomass' is not correct. This is because colored detrital organic materials and euphotic zone depth do not reflect biomass at the sea surface.

We agree and the text has been updated to:
    "We studied the relationship between marine aerosols measured over the North Atlantic
    Ocean during NAAMES and back trajectories weighted by four metrics of satellite
    measured ocean biological and physical properties (chlorophyll-a, sea water particulate
    organic carbon, colored detrital organic materials, euphotic zone depth), modelled net
    primary production, and model reanalysis meteorological parameters."
Line 440: replace 'abundance' with 'biomass'
Fixed
Line 447: What does 'surface biomass' indicate? 'Net primary production'? NPP does not mean biomass.
We have changed 'surface biomass' to 'net primary production'.

Provide a clear explanation for 'refractory' and 'non-refractory' particles.
We have updated the text to clearly state the difference:
     "The AMS does not efficiently measure refractory particles (i.e. particles that do not
     efficiently vaporize at 600°C), such as sea salt particles."

Lines 31-35 (abstract), 445-453 (conclusion), and relevant explanation in the Results and Discussion section: The explanation in these parts is confusing and hard to follow. As the authors noted, the lifetime of DMS in the atmosphere (1-2 days) is longer than that of other trace gases such as isoprene and monoterpene (less than a few hours). Considering the typical growth rate of SOA particles in the marine atmosphere, the difference in the lifetime of these VOCs (DMS, isoprene, etc.) may not significantly affect their temporal contribution to the organic aerosol mass over the study period. Moreover, the North Atlantic Ocean is well-known for high sea water DMS concentrations (a few nM, and occasionally increasing up to hundreds of nM during the phytoplankton bloom period due to the high abundance of DMS-producing phytoplankton groups such as haptophytes). The seawater concentration of isoprene (a few pM) in the North Atlantic Ocean is much lower than that of DMS (e.g., Dani and Loreto, 2017).

We agree that the explanation provided requires greater detail, particularly the complexity of DMS and its link (or lack of) to particle concentrations. We understand that the levels of isoprene in seawater are relatively low when compared to DMS concentration. However, models have suggested there may be an undiscovered source of VOCs that leads to the formation of SOA. Being an unknown source, we can only put our results in context of what is known. We have included this caveat in the introduction:

> "While isoprene and monoterpenes are known precursors for secondary organic particle mass, models indicate previously observed particle yields and estimated air-sea fluxes of isoprene (2%, 13–38 $\mu g\ m^{-2}d^{-1}$) and monoterpenes (~32%, 0.27–0.78 $\mu g\ m^{-2}d^{-1}$) (Hu et al., 2013; Lee et al., 2006) are too low to account for the observed MBL organic mass, suggesting that there may be large undiscovered sources (Arnold et al., 2009; Myriokefalitakis et al., 2010)."

We have also added the relevant updated text below to the results section:

> "DMS also has a number of chemical pathways with various secondary aerosol yields, making a direct link to biological processes more challenging (Faloona et al., 2009) "

And

> "Also, $SO_2$, a DMS oxidation product, has a lifetime on the order of days to weeks."

We have also added updated the below text in the conclusion:

> "Furthermore, the longer lifetime of DMS and its oxidation products can delay the formation of sulfate aerosol mass, making sulfate precursors more likely to advect through long-range transport if vertically lofted into the free troposphere and re-entrained down into the MBL. MBL to free troposphere transport of DMS is not captured well by the FLEXPART model. In addition, there are numerous DMS chemical pathways with various secondary aerosol yields that can obscure any link between sulfate aerosol concentrations and biogenic processes (Faloona et al., 2009)."

**Technical corrections**

Line 68: replace SO4 with SO42-
Fixed

Line 168: provide full name for SEMS

We have reordered sentences so the SEMS was introduced before this sentence in line 168.

**References**

Arnold, S.R., Spracklen, D.V., Gebhardt, S., Custer, T., Williams, J., Peeken, I., Alvain, S., 2010. Relationships between atmospheric organic compounds and air-mass exposure to marine biology. Environ. Chem. 7 (3), 232–241. https://doi.org/10.1071/EN09144.

Park, K.-T., Lee, K., Kim, T.-W., Yoon, Y.J., Jang, E.-H., Jang, S., Lee, B.- Y., Hermansen, O., 2018. Atmospheric DMS in the Arctic Ocean and Its Relation to Phytoplankton Biomass. Global Biogeochem. Cy. 32 (3), 351–359. https://doi.org/10.1002/2017GB005805.

Dani, K. G. S., and Loreto, F. (2017). Trade-off between dimethyl sulfide and isoprene emissions from marine phytoplankton. Trends Plant Sci. 22, 361–372. doi:10.1016/j.tplants.2017.01.006